# Machine Learning for ADHD Diagnosis: Feature Selection from Parent Reports, Self-Reports and Neuropsychological Measures

**DOI:** 10.3390/children12111448

**Published:** 2025-10-24

**Authors:** Yun-Wei Dai, Chia-Fen Hsu

**Affiliations:** Department of Occupational Therapy, College of Medicine, National Cheng Kung University, Tainan 701401, Taiwan

**Keywords:** attention-deficit/hyperactivity disorder, machine learning, feature selection, nested cross-validation, executive function, self-control, ex-Gaussian

## Abstract

**Highlights:**

**What are the main findings?**
Social problems, executive dysfunction, and self-regulation were top predictors of ADHD beyond core symptoms across machine learning models.Ex-Gaussian reaction-time parameters outperformed traditional indices of the continuous performance task.

**What is the implication of the main finding?**
Prioritizing these key predictors can streamline ADHD assessment and support earlier referral and intervention.Interpretable machine learning models can support clinical decision-making by highlighting the most informative features.

**Abstract:**

**Background:** Attention-deficit/hyperactivity disorder (ADHD) is a heterogeneous neurodevelopmental condition that currently relies on subjective clinical judgment for diagnosis, emphasizing the need for objective, clinically applicable tools. **Methods:** We applied machine learning techniques to parent reports, self-reports, and performance-based measures in a sample of 255 Taiwanese children and adolescents (108 ADHD and 147 controls; mean age = 11.85 years). Models were trained under a nested cross-validation framework to avoid performance overestimation. **Results:** Most models achieved high classification accuracy (AUCs ≈ 0.886–0.906), while convergent feature importance across models highlighted parent-rated social problems, executive dysfunction, and self-regulation traits as robust predictors. Additionally, ex-Gaussian parameters derived from reaction time distributions on the Continuous Performance Test (CPT) proved more informative than raw scores. **Conclusions:** These findings support the utility of integrating multi-informant ratings and task-based measures in interpretable ML models to enhance ADHD diagnosis in clinical practice.

## 1. Introduction

Attention-deficit/hyperactivity disorder (ADHD) is one of the most common neurodevelopmental disorders in children and adolescents, with global prevalence estimated to range from 3.4% to 8.0% [1,2,3]. The core symptoms of ADHD, i.e. inattention, hyperactivity and impulsivity, lead to significant functional impairments across different settings. In addition, children and adolescents with ADHD frequently exhibit deficits in executive functions, including inhibitory control [4] and working memory [5]. Beyond cognitive impairments, studies have also documented a wide range of adverse behavioral outcomes, including sleep problems [6], emotional dysregulation [7], and social difficulties [8]. These difficulties negatively affect peer and family relationships.

These wide-ranging impairments highlight the importance of comprehensive assessment for accurate and timely diagnosis. Yet, despite advances in understanding ADHD’s neurobiological basis, diagnosis continues to rely heavily on the clinician’s judgment. A comprehensive diagnostic process typically involves integrating multiple sources of information, including diagnostic interviews and standardized psychological assessments. Rating scales are among the most commonly used tools in clinical practice, as they enable efficient, multi-informant screening and symptom monitoring [9,10]. However, rating scales alone should not be used to make a diagnosis, as this may increase the risk of overdiagnosis [11]. Neuropsychological tests provide objective measures of core cognitive abilities such as working memory [12] and sustained attention [13]. Nevertheless, their feasibility in busy clinical settings is limited, as they require specialized software, trained personnel, considerable administration time, and are costly [14].

Given these challenges, the unique contribution of the present study is to systematically integrate multi-informant ratings and performance-based measures. We also rely on interpretable machine learning (ML) techniques, with a specific focus on identifying consensus predictors across models.

Machine learning is particularly well-suited to this challenge because ADHD is a heterogeneous disorder influenced by multiple interacting factors [15]. In contrast, traditional linear approaches fail to accommodate the non-linear, multidimensional nature of ADHD [16,17]. Recent studies have demonstrated promising predictive performance across various ML algorithms. For instance, Liu et al. [18] developed ML models using administrative health data and teacher ratings to prospectively predict ADHD. Mikolas et al. [19] employed a support vector machine (SVM) with automatic feature selection, achieving an AUC of 68.1% and emphasizing the value of integrating symptom measures across domains. Other work has gone beyond prediction to explore feature importance. Christiansen et al. [20] classified ADHD and other conditions using logistic regression (accuracy = 0.79), SVM (accuracy = 0.82) and LightGBM (accuracy = 0.80). They also reported item-specific feature importance of Conner’s Adult ADHD Rating Scales. Garcia-Argibay et al. [21] compared several ML models to identify consensus predictors, such as gender, family history, and academic difficulties. Neural networks have also been used to classify ADHD subtypes based on rating scale and performance test data, with improved accuracy observed when newer versions of continuous performance tasks were applied [22]. Collectively, these studies illustrate that in addition to prediction, ML approaches offer interpretable insights through feature importance measures, helping to identify which scales, subscales, neuropsychological measures, and demographic factors most strongly contribute to classification accuracy [15].

Together, these studies show that ML models predict ADHD with high accuracy. However, the majority of research has focused on prediction accuracy without addressing interpretability. This limits clinical relevance because model accuracy alone does not guide the selection of meaningful predictors. Furthermore, some studies evaluate without a held-out test set, potentially leading to overestimation [23,24].

To address these gaps, we designed the present study using a broad dataset from two prior ADHD research projects. We employed a comprehensive set of measures, including commonly used clinical tools for assessing ADHD, as well as measures of ADHD-related characteristics such as executive function and proneness to boredom. These measures were drawn from self-reports, parent reports and performance-based tasks. This wide range of measurements provided a unique opportunity to simultaneously compare diverse indicators, including tools typically used in research contexts.

Our primary objective is to identify the most predictive features for ADHD diagnosis. To this end, we systematically compare a range of ML model families that capture different characteristics of the data. This includes regularized regression for feature selection and transparency, SVM for capturing non-linear patterns, decision trees for rule-based interpretability, and ensemble methods for robust predictive performance. To mitigate the risk of overestimating model performance, we employed nested cross-validation [25] (Steinert et al., 2024), which separates hyperparameter tuning from performance evaluation. We aim to demonstrate how ML can improve predictive accuracy while also supporting the refinement of clinically practical assessment tools.

## 2. Materials and Methods

### 2.1. Participants

The dataset comprised 255 participants drawn from two research projects investigating cognitive function and emotional dysregulation in ADHD (Chang Gung Medical Foundation Institutional Review Board ref. 201701995B0 and 201902080B0). This combined sample included 108 children with ADHD and 147 typically developing controls, all aged between 9 and 16 years. Participants were recruited from Chang Gung Memorial Hospitals and local communities in North and Central Taiwan. ADHD diagnoses were confirmed by senior child psychiatrists based on criteria from the Diagnostic and Statistical Manual of Mental Disorders, Fifth Edition (DSM-5). Exclusion criteria included (a) the presence of neurodevelopmental or psychiatric disorders other than ADD/ADHD; (b) an estimated IQ below 70, and (c) the use of medication other than short-acting stimulants. Individuals with ADHD were instructed to refrain from taking medication on the day prior to testing. All participants completed Conners’ Continuous Performance Test (CPT), the short form of the Wechsler Intelligence Scale for Children, 4th Edition (WISC-IV) [26], and self-ratings assessing delay aversion, boredom, and mind-wandering. Parents or primary caregivers completed parent-rating scales to assess ADHD-related symptoms, emotional and behavioral problems.

### 2.2. Measures

We included predictors across four domains commonly implicated in ADHD.

#### 2.2.1. Demographics

We collected demographic information, including participants’ age, years in education, gender, and handedness. Additional variables were also recorded, including hours of sleep, hours of sleep for the previous night, reported sleep problems, school performance, history of preterm birth, epilepsy, and head trauma.

#### 2.2.2. Cognitive Task Performance

All participants completed four subtests of the WISC-IV (Similarities, Matrix Reasoning, Digit Span, and Symbol Search) for estimated IQ. This specific subset has been shown to highly correlate with full-scale IQ in Taiwanese samples [27].

In addition, participants completed Conners’ Continuous Performance Test (CPT-3) [28], a tool commonly used to assess the sustained attention and impulsivity of children at risk for ADHD. From CPT-3, we extracted the raw scores of standard performance indices. We also fit ex-Gaussian distributions to the RT data, deriving the μ, σ, and τ parameters, as these have been shown to effectively differentiate individuals with ADHD from controls [29,30].

#### 2.2.3. Self and Parent Ratings

Participants completed a set of questionnaires assessing spontaneous mind-wandering, trait and state boredom, delay-aversion/discounting, and self-control. These included the Mind-Wandering: Spontaneous Scale (MWS) [31], the Short Boredom Proneness Scale (SBPS [32]; adaptation to Traditional Chinese [33]), the Multidimensional State Boredom Scale (MSBS-8) [34], Quick Delay Questionnaire (QDQ) [35], and the Brief Self-Control Scale (BSCS) [36].

A parent or guardian completed the parent version of the aforementioned questionnaires. Additionally, they completed the Childhood Executive Function Inventory (CHEXI) [37] to assess their child’s daily executive functioning difficulties and the Child Behavior Checklist (CBCL) [38] to evaluate internalizing and externalizing behavioral problems. For the CBCL, the Attention Problem subscale and the DSM-oriented scales were excluded due to their strong correlation with ADHD diagnosis, which could unduly dominate predictive models. Parents also completed the Swanson, Nolan, and Pelham Rating Scale (SNAP-IV) [39] to assess ADHD symptom severity; however, this measure was likewise excluded from analysis for the same reason.

A summary of all indices used in the analysis is presented in Table 1.

### 2.3. Data Preparation

All analyses were performed in R version 4.3.2 using tidymodels (v1.3.0 [40]). In each resampling split, preprocessing was applied to all data: mean-imputation for numeric variables, dummy coding for nominal predictors, and removal of zero-variance variables. For linear and kernel models, numeric variables were standardized.

### 2.4. Models

We compared six supervised classifiers representing complementary inductive biases.

LASSO and Elastic Net regularized logistic regression models were fit using the glmnet package (v4.1-8) with penalty strength (and the mixing parameter for Elastic Net) tuned over log-scaled grids.An SVM with radial basis function(RBF) kernel was tuned over log-scaled grids of cost and kernel width using the kernlab engine (v0.9.33).Random Forests (ranger, v0.17.0) were tuned on the number of predictors sampled at each split, while the number of trees was fixed at 300.Gradient-boosted trees were fit using xgboost (v1.7.11.1) with a logistic objective; tree count was fixed at 300, and tuning was performed for maximum depth and learning rate.A single decision tree (rpart, v4.1.23) was tuned on the cost-complexity pruning parameter and maximum depth.

### 2.5. Nested Cross-Validation

We employed nested cross-validation (CV) to separate hyperparameter tuning from performance estimation. The outer loop consisted of a 5-fold stratified CV based on ADHD diagnosis. Within each outer training split, hyperparameters were selected using an inner 5-fold CV with the area under the receiver operating characteristic curve (ROC AUC) as the tuning criterion.

The tuned model was evaluated on the held-out outer test split to provide an unbiased estimate of performance. Specifically, accuracy, F1 score, and AUC were averaged across the outer folds for reporting.

### 2.6. Model-Specific Feature Importance

Model-specific feature importance measures were extracted using the APIs provided by each learning engine.

For LASSO and Elastic Net, absolute standardized coefficients from tuned fits on the full dataset (with the same preprocessing procedure) were used as direct measures of feature importance.For the SVM with RBF kernel, permutation importance was estimated as the drop in AUC.Random Forest and decision tree models provided impurity-based importances, which were averaged across outer folds.For XGBoost, gain-based importances were computed within each outer fold and then averaged.

### 2.7. Model-Agnostic Feature Importance

To complement the model-specific feature importance rankings, we computed Shapley additive explanation (SHAP) values to provide a model-agnostic measure of feature contributions. SHAP values estimate the marginal contribution of each feature to a model’s prediction. Positive SHAP values indicate that a feature increases the predicted probability of ADHD relative to the baseline, whereas negative values indicate a decrease. Because SHAP values do not rely on model-specific feature importance indices (e.g., impurity reduction in trees or regression coefficients), they provide a model-agnostic scale of feature contributions.

For each model, SHAP values were computed on the held-out test split of each outer fold (fastshap, v0.1.1 [41]). We then averaged mean absolute SHAP values across folds to provide an interpretable summary of feature contributions to out-of-sample predictions.

## 3. Results

### 3.1. Sample Characteristics

Table 2 summarizes participant characteristics by group. The control group was slightly older than the ADHD group (Welch two-sample *t*-test, *t*(237.70) = 1.96, *p* = .051, 95% CI [0.00, 0.89], d = 0.25). Gender distribution differed significantly (Fisher’s exact test, 82% vs. 62%, OR = 2.87, *p* < .001), with a higher proportion of males in the ADHD group.

### 3.2. Model Performance

Fitted model AUC, Accuracies, and F1-Scores are averaged across outer folds in five-fold nested cross-validation (Table 3).

The performance of the two regularized logistic regression models was similar, where LASSO achieved the best performance with the highest mean AUC (0.898 ± 0.052), alongside an accuracy of 0.799 ± 0.067 and F1-Score of 0.755 ± 0.076. The SVM with non-linear kernel achieved a respectable mean AUC (0.897 ± 0.055), accuracy (0.804 ± 0.068) and F1-Score (0.764 ± 0.081).

Predictive performance was lower for the single decision tree CART method (AUC = 0.747 ± 0.038; accuracy = 0.761 ± 0.028; F1 = 0.690 ± 0.041) whereas ensemble tree methods achieved substantially better results. Specifically, Random Forest achieved an AUC of 0.900 ± 0.034, accuracy of 0.808 ± 0.050, and F1-Score of 0.767 ± 0.045. The best fitting model was XGBoost, achieving an AUC of 0.906 ± 0.032, accuracy of 0.824 ± 0.023, and F1-Score of 0.787 ± 0.019.

ROC curves for all six models (Figure 1) showed substantial overlap except for the single decision-tree, corroborating the quantitative differences reported in Table 3.

### 3.3. Model-Specific Feature Contributions

Table 4 summarizes the top 30 predictors of the ADHD classification based on native importance metrics across six machine learning models. Despite differences in algorithmic mechanisms, several predictors consistently emerged as highly influential. Parent-reported social problems from the CBCL were among the strongest predictors, reflecting its prominence as a composite indicator of behavioral impairment. Spontaneous mind-wandering ratings and trait self-control (BSCS) also ranked highly, pointing to self-regulation traits that are often overlooked in routine clinical assessment. Executive function deficits, as measured by the CHEXI, were similarly robust predictors, aligning with the cognitive challenges central to ADHD. Notably, some of these measures, such as the MWS and BSCS, are not typically included in standard ADHD evaluations. Our findings suggest that incorporating these tools could enhance clinical assessment.

Computerized performance measures from Conners’ Continuous Performance Test (CPT) also emerged as relevant predictors. Notably, ex-Gaussian parameters (τ, σ) were preferred by most models, suggesting that distributional characteristics of reaction time performance contribute more than raw indices.

### 3.4. Model-Agnostic Feature Contributions

The model-agnostic importance was estimated using SHAP values (Figure 2). The global SHAP summary, averaged across outer folds, confirmed the convergence observed in native importance metrics. The most influential predictors across the models were social problems (CBCL), working memory (CHEXI), parent reports of self-control (BSCS), inhibition (CHEXI), and spontaneous mind-wandering (MWS), all of which exhibited the largest mean SHAP magnitudes.

## 4. Discussion

The diagnosis of ADHD relies on subjective clinical judgment based on information from multiple sources, which are often highly intercorrelated [42,43,44]. While univariate analyses can evaluate the contribution of individual variables, they risk overlooking critical interactions among them [16,17]. The current study employed ML models capable of tackling the following challenge: regularized linear models that select features while managing collinearity, and non-linear models that capture complex interactions. Six machine learning models were trained with a nested cross-validation framework, and all achieved satisfactory AUCs except for the decision tree algorithm. These findings suggest that combining parent-report, self-report and neuropsychological measures can effectively classify ADHD.

The robust performance of linear and kernel-based models in our study supports a dimensional rather than categorical view of ADHD. Consistent with findings by [45], symptom severity appears to vary continuously between individuals rather than forming discrete clusters. This helps explain why tree-based algorithms, which rely on hard splits, performed less effectively. This aligns with the broader conceptual shift in ADHD classification. The DSM-5 replaced the notion of stable, trait-like subtypes with presentations, reflecting growing evidence that ADHD symptom profiles are dynamic and evolving across development rather than fixed categories [46]. Thus, the dimensional framework of our findings mirrors both empirical trends and clinical observations.

Across models, the CBCL social problems scale consistently ranked among the most important predictors, underscoring the social difficulties that often accompany functional impairment [47]. Such social difficulties may stem from executive dysfunction [48] and emotional dysregulation [49], suggesting that the social domain represents a composite measure of multiple origins. Consistent with this mechanism, inhibition and working memory scores from the CHEXI contributed substantially to prediction. These processes have long been recognized as core executive dysfunctions in ADHD [50,51].

Furthermore, self-regulatory capacity emerged as a robust predictor. Consistent with prior findings, measures of self-control correlated strongly with ADHD symptom severity [52,53]. Another aspect of self-regulation stemming from internal distractibility has been conceptualized as mind-wandering [54]. Empirical studies show how spontaneous mind-wandering correlates with greater ADHD symptom severity. In the present study, both BSCS(self-control) and MWS(spontaneous mind-wandering) scores emerged as robust predictors, highlighting their potential as useful clinical assessment tools.

Nevertheless, the reliability of introspective ratings can vary with cognitive maturity [55], self-consciousness [56], and reporting bias [57,58,59]. Indeed, previous studies have suggested that self-reported measures are generally less predictive than parent reports [60,61]. Interestingly, our models ranked self-reports of the delay discounting scale on QDQ relatively high, suggesting that such measures may still provide useful complementary information in clinical assessment.

The Continuous Performance Test (CPT) has been proposed as an objective measure for ADHD diagnosis [62], though evidence has cast doubt on its diagnostic value [63]. Computational modeling approaches have emphasized modeling the shape of the reaction time distribution instead of raw scores [29,30]. In current research, we discovered that ex-Gaussian indices were preferred by the models, suggesting that subtle attentional lapses may be better quantified through computational modeling.

Taken together, our findings indicate that social difficulties (CBCL social problems), executive dysfunction (CHEXI), and self-regulation difficulties (BSCS, MWS) consistently predict ADHD across models. These domains should be prioritized in clinical prescreening to improve diagnostic speed and accuracy. Some of the scales have teacher-rated versions, which may offer a practical tool for early identification and timely referral.

Despite these contributions, the current study is limited by a lack of out-of-distribution test cases and a relatively small sample size drawn from a single country. However, the participants were recruited from urban and suburban areas in Northern and Central Taiwan to help reduce potential socioeconomic bias. To avoid inflating performance estimates [23,24,64], we adopted a nested cross-validation approach, in which an outer loop was used to evaluate model performance independently of hyperparameter optimization. This nested design reduced the risk of overestimation for predictive power. However, we emphasize that the reported performance should be regarded primarily as a demonstration of methodological validity. The focus of this study is on feature selection with interpretable machine learning techniques, and the generalizability of predictive accuracy beyond the current sample remains to be established. Future research should involve larger, more socioeconomically and culturally diverse samples to validate the stability of the predictors identified.

## 5. Conclusions

This study demonstrated that integrating self-reports, parent reports, and neuropsychological tests within a machine learning framework can support interpretable clinical decision-making. Across six diverse models, consensus feature rankings consistently highlighted social problems (CBCL), executive dysfunction (CHEXI), and self-regulation difficulties (BSCS, MWS) as key predictors. Additionally, ex-Gaussian indices derived from computational modeling of reaction time distributions provided greater classification utility than raw metrics, underscoring the value of computational modeling in enhancing computerized assessments. Taken together, these findings support the use of machine learning as an interpretable approach to feature selection in clinical and academic settings. Future research should validate these predictors in larger, out-of-sample cohorts and explore their translation into practical assessment routines.

## Figures and Tables

**Figure 1 children-12-01448-f001:**
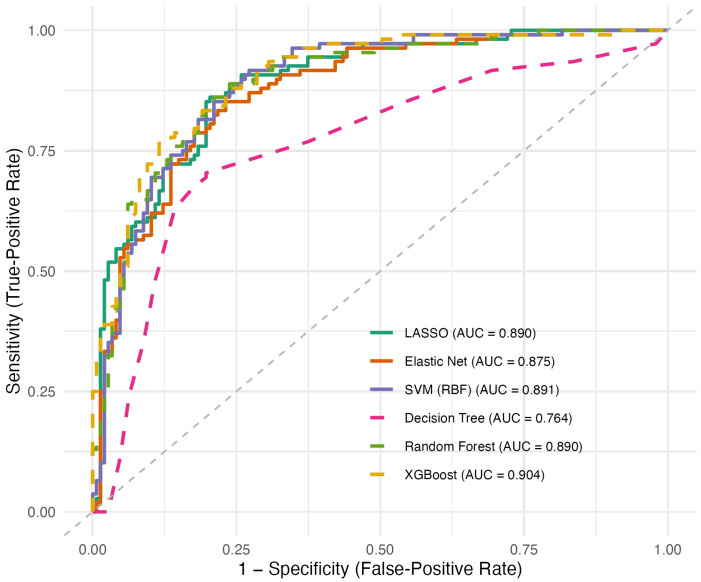
ROC curves for fitted models.

**Figure 2 children-12-01448-f002:**
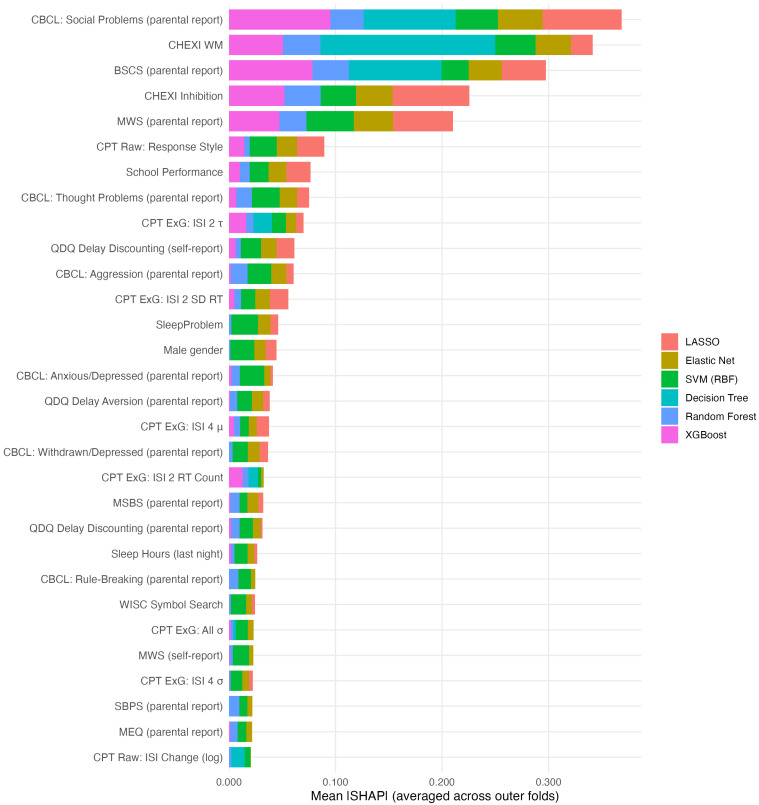
Mean absolute SHAP values for top 30 features. Note. BSCS: Brief Self-Control Scale; CBCL: Child Behavior Checklist; CHEXI: Childhood Executive Function Inventory; CPT: Conners’ Continuous Performance Test; MWS: Mind-Wandering: Spontaneous Scale; QDQ: Quick Delay Questionnaire.

**Table 1 children-12-01448-t001:** Summary of indices used in the analysis.

Domain	Tools	Variables
Demographics	Demographic questionnaire	Age, Education, Gender, Handedness, and School performance; Sleep hours (last night), Sleep hours (typical night), and Sleep problems; History of preterm birth, epilepsy or head trauma.
Cognitive Performance	Wechsler Intelligence Scale for Children, 4th Edition (WISC-IV)	Scaled scores on Digit Span, Matrix Reasoning, Similarities, Symbol Search, and Estimated IQ.
	Conners’ Continuous Performance Test (CPT)	CPT Raw scores Response Style, Hits, $d^{\prime }$, Omissions, Commissions, and Perseverations. Hit Reaction Time (HRT) and Hit Reaction Time Standard Deviation (HRT SD), Variability, HRT Block Change and HRT Inter-Stimulus Interval (ISI) Change, analyzed in both raw and log-transformed form. ISI Change: Change in RT across ISIs.
		CPT ex-Gaussian parameters Mean RT (all trials), RT Count (total number of RT observations included across all trials), and SD RT. μ (Gaussian mean component across all trials). σ (Gaussian SD component across all trials). τ (exponential tail). The ex-Gaussian metrics are computed for all trials, including the first half of all trials, last half of all trials, and across ISIs of 1, 2 and 4 s. Variable names follow the convention: CPT ExG 〈condition〉 〈metric〉. e.g., CPT ExG: ISI 1 μ.
Self-ratings	Brief Self-Control Scale (BSCS), Quick Delay Questionnaire (QDQ), Multidimensional State Boredom Scale (MSBS), Short Boredom Proneness Scale (SBPS), and Mind-Wandering: Spontaneous Scale (MWS)	Quick Delay Questionnaire (QDQ) includes two subscales: delay aversion and delay discounting. For all other measures, total scores were used.
Parent-ratings	Childhood Executive Function Inventory (CHEXI), Child Behavior Checklist (CBCL), BSCS, QDQ, MSBS, SBPS, and MWS	CHEXI subscales, including Inhibition and Working Memory (WM). CBCL subscales, including Aggression, Anxious/Depressed, and Rule-Breaking. Social Problems, Somatic Complaints, Thought Problems, and Withdrawn/Depressed. For all other measures, total scores were used.

**Table 2 children-12-01448-t002:** Sample characteristics by group.

Characteristic	Overall	Control	ADHD	Statistic	*p*-Value
	N = 255	N = 147	N = 108	t/OR	
Age	11.85 (1.80)	12.03 (1.84)	11.59 (1.73)	1.96	.051
Gender				2.87	<.001
Female	75 (29%)	56 (38%)	19 (18%)		
Male	180 (71%)	91 (62%)	89 (82%)		
Handedness				0.86	.829
Left	24 (9.4%)	13 (8.8%)	11 (10%)		
Right	231 (91%)	134 (91%)	97 (90%)		

**Table 3 children-12-01448-t003:** Model performance in nested cross-validation for ADHD classification.

Model	Accuracy	AUC	F1
LASSO	0.799 (0.067)	0.898 (0.052)	0.755 (0.076)
Elastic Net	0.799 (0.071)	0.886 (0.056)	0.755 (0.081)
SVM(RBF)	0.804 (0.068)	0.897 (0.055)	0.764 (0.081)
Decision Tree	0.761 (0.028)	0.747 (0.038)	0.690 (0.041)
Random Forest	0.808 (0.050)	0.900 (0.034)	0.767 (0.045)
**XGBoost**	**0.824 (0.023)**	**0.906 (0.032)**	**0.787 (0.019)**

Note. Values represent the mean (standard deviation) across outer cross-validation folds. AUC = area under the ROC curve. SVM: support vector machine.

**Table 4 children-12-01448-t004:** Native feature-importance ranks for top 30 features across models.

**Feature**	LASSO	Elastic Net	SVM (RBF)	Random Forest	XGBoost	Decision Tree
CBCL Syndrome: Social Problems (parent-reported)	2	1	2	3	1	4
CHEXI Inhibition Total	1	2	14	1	4	2
MWS Total (parent-reported)	3	3	1	5	5	9
CHEXI WM Total	7	5	17	2	3	1
BSCS Total (parent-reported)	4	4	32	4	2	3
CPT Raw: Response Style	9	6	6	17	11	—
School Performance	5	7	31	12	10	—
CBCL Syndrome: Thought Problems (parent-reported)	10	9	28	7	18	10
CPT ExG: ISI 2 SD RT	6	10	23	19	19	17
QDQ Delay Discounting (self-reported)	8	8	16	20	27	41
CPT ExG: ISI 2 τ	64	13	19	24	8	13
Delay Aversion (self-reported)	22	36	38	27	16	—
CBCL Syndrome: Anxious/Depressed (parent-reported)	80	14	15	15	33	12
WISC Similarities (SS)	13	28	20	54	28	—
MWS Total (self-reported)	19	33	13	34	47	—
QDQ Delay Discounting (parent-reported)	78	19	22	10	41	7
CBCL Syndrome: Aggression (parent-reported)	84	18	35	6	30	5
CPT ExG: All σ	39	16	10	65	26	23
CPT Raw: Hits	24	38	50	32	22	22
CPT ExG: ISI 1 τ	58	23	12	57	17	—
SBPS Total (self-reported)	21	35	59	36	20	36
CPT ExG: ISI 4 μ	67	17	55	26	13	—
CPT Raw: Block Change (log)	35	47	37	30	29	—
CPT ExG: All RT Count	41	52	69	18	15	20
CPT Raw: RT SD (log)	31	44	66	16	23	—
CPT Raw: ISI Change (log)	37	49	42	37	—	16
CPT Raw: Omissions	26	20	52	62	—	27
CPT Raw: Hit RT	29	42	18	45	54	—
CPT Raw: Perseverations	28	41	43	55	21	—
WISC Digit Span (SS)	14	29	25	68	56	37

Note. Cells show within-model ranking (1 = most important). “—” indicates that the feature was not ranked by that model. Rows are ordered by average rank over available models (not displayed). BSCS: Brief Self-Control Scale; CBCL: Child Behavior Checklist; CHEXI: Childhood Executive Function Inventory; CPT: Conners’ Continuous Performance Test; MWS: Mind-Wandering: Spontaneous Scale; QDQ: Quick Delay Questionnaire; WISC: Wechsler Intelligence Scale for Children.

## Data Availability

The raw data supporting the conclusions of this article will be made available by the authors on request.

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
