# Peer review of "Machine Learning for ADHD Diagnosis: Feature Selection from Parent Reports, Self-Reports and Neuropsychological Measures"

_children, 2025, doi:10.3390/children12111448_

Round 1

Reviewer 1 Report

Comments and Suggestions for Authors

Dear authors,

Thank you for allowing me to read your manuscript “Machine Learning for ADHD Diagnosis: Feature Selection from Parent/Self-Reports and Neuropsychological Measures". This study could be relevant; I have only minor recommendations.

Title

The title is clear and descriptive.

“Parent/Self-Reports” may be unclear for readers unfamiliar with ADHD research (is it parent reports, self-reports, or both?).

Abstract

Quite dense for an abstract; some details (e.g., all model names, standard deviations) may be better in the main text.

Introduction

While gaps are noted (interpretability, risk of overestimation), the unique contribution of this study should be highlighted earlier and more strongly.

Materials and methods

The lists of CPT variables, subscale names, and questionnaire items may overwhelm readers—these could be summarized in a main text table (with details in an appendix).

Results

Highlight why certain features (e.g., social problems, mind-wandering, executive function deficits) are clinically meaningful.

Report standardized effect sizes (e.g., Cohen’s d for age) where possible.

For categorical variables, present proportions alongside ORs to aid interpretation.

Discussion

Clinical utility is mentioned, but not deeply explored (e.g., how could these findings change assessment protocols in clinics or schools?).

Mentions sample size and lack of out-of-distribution testing but doesn’t address cultural specificity (all participants Taiwanese), or exclusion of ADHD subtypes.

Conclusion

No comments

References Well-referenced.

Regards,

Reviewer 2 Report

Comments and Suggestions for Authors

Thank you for the opportunity to review this manuscript. The manuscript addresses an important topic and has many merits. However, I have some comments and concerns.

1) Some sentences are overly long or dense, which may hinder readability. For example, ML models can analyze complex patterns across multi-informant reports and neuropsychological test performance, offering support for clinical decision-making and uncovering features that may not be readily apparent through traditional methods. Consider breaking this into two sentences for clarity.

2) The phrase “self-/parent reported measures” could be clarified as “self-report and parent-report measures” for smoother reading.

3) The authors clearly articulate the aim of your study, but the novelty of your dataset is not yet fully emphasized.

4)The flow between some paragraphs could be improved. For example, the transition from discussing the general challenges of diagnosis to introducing machine learning could be smoother.

5)  SHAP is mentioned as a model-agnostic method, which is excellent. A brief explanation of what SHAP values represent (e.g., contribution of each feature to prediction) would help readers unfamiliar with the concept.

6) The authors state that analyses were performed in R (tidymodels). That’s good, but it would strengthen reproducibility to mention exact version numbers of R and packages (tidymodels, glmnet, xgboost, etc.).

7) Section 2.4–2.6 (Models, CV, Feature importance) is technically correct but very condensed. Breaking into shorter sentences and subpoints would improve clarity.

8) The discussion reads like a literature mini-review rather than a flowing narrative. Many citations are clustered without enough interpretive synthesis.

9) The authors highlight predictors (social problems, executive dysfunction, self-regulation, ex-Gaussian indices), but the discussion doesn’t fully explore how these can change assessment or practice.

10) Sample limitation underdeveloped: The authors acknowledge small sample size but don’t discuss potential biases (e.g., single-country, hospital/community recruitment, possible socioeconomic skew).

11) The discussion of self-report measures could benefit from a more balanced tone. While introspective ratings may capture internal states, it’s worth noting that their reliability can vary by age and cognitive maturity.

Round 2

Reviewer 2 Report

Comments and Suggestions for Authors

I would like to thank the authors for making the requested changes accordingly. The manuscript has improved greatly.